# Detecting causality signal in instrumental measurements and climate model simulations: global warming case study

*Mikhail Y. Verbitsky[1]* [,][a], Michael E. Mann[2], Byron A. Steinman[3], Dmitry M. Volobuev[4]*

[1]Gen5 Group, LLC, Newton, MA, USA
[2]Department of Meteorology, The Pennsylvania State University, University Park, PA, USA
[3]Large Lakes Observatory and Department of Earth and Environmental Sciences, University of Minnesota Duluth, Duluth, MN, USA
[4]The Central Astronomical Observatory of the Russian Academy of Sciences at Pulkovo, Saint Petersburg, Russia
*retired
[a]formerly at: Yale University, Department of Geology and Geophysics, New Haven, CT, USA
Correspondence: Mikhail Verbitsky (verbitskys@gmail.com)

**Abstract.** Detecting the direction and strength of the causality signal in observed time series is becoming a popular tool for exploration of distributed systems such as Earth's climate system. Here we suggest that in addition to reproducing observed time series of climate variables within required accuracy a model should also exhibit the causality relationship between variables found in nature. Specifically, we propose a novel framework for a comprehensive analysis of climate model responses to external natural and anthropogenic forcing based on the method of conditional dispersion. As an illustration, we assess the causal relationship between anthropogenic forcing (i.e., atmospheric carbon dioxide concentration) and surface temperature anomalies. We demonstrate a strong directional causality between global temperatures and carbon dioxide concentrations (meaning that carbon dioxide affects temperature stronger than temperature affects carbon dioxide) in both the observations and in (CMIP5) climate model simulated temperatures.

## 1. Introduction

The standard approach to attribution of observed global warming employs experiments with climate models. Such "detection and attribution" approaches (e.g., Stocker, 2014) attempt to reproduce observed trends under different external forcing conditions and demonstrate a consistency (or its absence) of simulated climate changes with instrumental observations. A substantial body of detection and attribution studies (e.g., Santer, *et al*., 2009, 2012; Jones, *et al*., 2013) spanning the past two decades demonstrates that anthropogenic increases in atmospheric carbon dioxide are very likely the cause of the observed global temperature increase since the mid-19[th] century. Semi-empirical approaches that combine information from model simulations and observations have also proven useful for investigations of modern climate change attribution. Previous work (e.g., Mann *et al*., 2017) has employed estimates of natural variability derived from a combination of historical simulations and observations to attribute the sequence of record-breaking global temperatures in 2014, 2015, and 2016 to anthropogenic warming by demonstrating that this sequence had a negligible likelihood of occurrence in the absence of anthropogenic warming. Direct investigations of the causal relationship between climate system variables using statistical tools have recently become more common. The most simplistic approach, the Pearson's correlation between two time series, which is often mentioned in the context of causality, does not really measure the causality. While statistically significant correlation quantifies similarity between time series, it does not imply a causality resulting from physical relationships between the natural processes that are expressed by the time series and that can be modeled using differential equations. Instead, it provides a statistical test of a hypothesis that describes a physical link between the two variables (i.e., expressed as times series) without actually testing neither the direction of causality nor the plausibility of the physics underlying the hypothesis. The breakthrough Granger developments (Granger, 1969) provided a foundation for several causality measuring techniques based on different hypotheses of data origin. The requirement of the cause leading the effect (but not *vice-versa*) defines the direction of a causal link if a more general hypothesis of lagged linear connection between noisy autoregressive processes is assumed. Though this hypothesis leads to statistically significant estimates of climate response to the forcing input (e.g., Kaufmann et al., 2006, 2011, Attanasio, 2012, Attanasio et al., 2012, Mokhov et al., 2012, Triacca et al., 2013), it may not be able to reliably detect the direction of causality in the climate system because the potential for non-linearities in the climate system (leading to extreme sensitivity to initial conditions, i.e., deterministic chaos) is not taken into account. For example, Palus et al. (2018) demonstrated that coupled chaotic dynamical systems can "violate the first principle of Granger causality that the cause precedes the effect." The Shannon information flow approach expands Granger causality to non-linear systems, using transfer entropy as a causality measure. Barnett *et al.* (2009) have shown that transfer entropy is equivalent to Granger causality for Gaussian processes. The transfer entropy between two probability distributions is typically considered the most general approach for causality detection, and numerous

modifications of transfer-entropy-based causality measuring techniques have been developed for different applications (Pearl, 2009), including causality measurements of global warming (e.g., Stips *et al*., 2016). It should be noted though that all probability based causality measures require long time series to calculate statistical distributions and may lack applicability to local climate due to high inhomogeneity and non-stationarity of the data (e.g., O'Brien et al., 2019). The prediction improvement approach is often considered as a generalization of Granger causality for non-linear systems (e.g., Krakovska and Hanzely, 2016). It is highly practical and, besides causality calculations, it may help to improve the prediction accuracy. For pure causality purposes, however, it adds an additional uncertainty because the causality may depend on the chosen prediction method. The convergent cross-mapping approach (Sugihara *et al*., 2012, Van Nes *et al*., 2015) has been recently designed to work with relatively short data series, thus addressing the major constraint of transfer-entropy approach. The background hypotheses of the method is more narrow and includes only nonlinear *dynamical* systems, though convergent cross-mapping remains applicable to most natural systems in ecology and geosciences (Sugihara *et al*., 2012). The approach considers conditional evolution of nearest neighbors in the reconstructed Takens' space so, it is sensitive to the noise and may not be applicable to a wide range of time scales. Moreover, Palus *et al.* (2018) have shown that convergent cross-mapping is not capable of determining the directionality of a causal link. Therefore identification of specific causal effect measures for climate observables is still a challenge. When causal effect measures are identified, the graph theory could be employed for further analysis of multiple causality chains (Hannart et al., 2016, Runge, et al., 2015). Along with dimensionality reduction formalism (e.g., Vejmelka et al., 2015) it may lead to a promising general approach.

For our case study, we advocate the method of conditional dispersion (MCD) developed by Čenys *et al*. (1991) as a causal effect measure. It has also been designed for non-linear systems and exploits the asymmetry of the conditional dispersion of two variables in Takens' space along all available scales. Therefore, it remains more general and noise resistant than convergent cross-mapping techniques and more general than prediction improvement approaches because it is insensitive to the choice of the prediction method. We propose here to employ the MCD-based causality measurements for a comprehensive analysis of climate model responses to external natural and anthropogenic forcing. While climate models have, in a rough sense, been tuned to reproduce the observational record, their predictions differ from the observations due to various types of errors and uncertainties. These include: (a) measurement errors in external forcing (e.g., greenhouse gas concentrations, land use, solar variability) used to drive the models; (b) errors in the representation of physical processes in the models (e.g., ocean circulation, cryosphere and biosphere processes, various feedback mechanisms, *etc.*) and incomplete representation of the Earth system (i.e., in many cases a lack of representations of dynamic vegetation responses, or the oceanic carbon cycle); (c) errors associated with internal variability in the climate system – for example, models may accurately represent ENSO, The El Niño-Southern Oscillation, but ENSO is an inherently random process and models therefore do not, and should not, reproduce the actual real-world realization of that random process; (d) errors and uncertainties in observational data - for example, surface temperature measurements contain uncertainty due to the irregular sampling in space and time (e.g., lack of data at higher latitudes increasingly back in time). In addition, there is the potential for biases due, for example, to changes in oceanic and terrestrial measurement platforms over time (e.g., bucket measurements *vs.* intake valves for ocean seawater measurements, or residual urban heat island biases in land-based temperature measurements). Such sampling uncertainties might lead to model – observational data mismatch that is unrelated to model performance. The challenge, then, is to determine the best-performing models when all reproduce the observations similarly well. We believe that in addition to reproducing observed time series of climate variables, a model should exhibit the causality relationship between variables found in nature. Since the MCD approach is based on the assumption that each time series is produced by a hypothetical low-dimensional system of dynamical equations, *similarity of causal relationships in both model and observations speaks to the similarities of their parent systems.*

Accordingly, our paper is structured as follows. First, we will briefly describe the method of conditional dispersion. We will illustrate it with several numerical experiments that investigate the causal relationship between surface temperature anomalies for the Northern Hemisphere and atmospheric carbon dioxide concentration measurements. We will show that the causality between carbon dioxide and temperature anomalies is a directional causality, meaning that carbon dioxide affects temperature stronger than temperature affects carbon dioxide. We will then demonstrate that this directional connection cannot be replicated with an independent trend and red noise.

## 2. A glimpse into the method of conditional dispersion.

The MCD approach has been designed for measuring causality between two time series. It is assumed that each time series is a variable produced by its hypothetical low-dimensional system of dynamical equations. The variables contain information about the dynamics of hypothetical parent systems which can be reconstructed using Takens

(1981) procedure. Since each of the variables can be used to reconstruct the original parent system manifold, there is one-to-one correspondence between them. Specifically, if points of one time series are close, the synchronous points of another variable are close as well. Therefore, if two variables ($u$ and $x$) do not belong to the same or coupled dynamical systems, or in other words, they are independent, then the distance from a reference point to its neighbors of one variable ($u$) does not depend on the distance ($\varepsilon$) between synchronous points of another variable ($x$). In the case of dependency, though, the distance between neighboring points of the controllable variable will be smaller when the distance between points of the driving variable is reduced. Therefore, the dependence of the conditional dispersion $\sigma(\varepsilon)$ of the variable $u$ upon the distance $\varepsilon$ between points of the variable $x$ becomes a signature of causal relationship between $u$ and $x$ (Čenys *et al.*, 1991):

$$\sigma_{xu}^{M}(\varepsilon) = \left( \frac{\sum_{i \neq j} \left\| u_i^M - u_j^M \right\|^2 \Theta(\varepsilon - \left\| x_i^M - x_j^M \right\|)}{\sum_{i \neq j} \Theta(\varepsilon - \left\| x_i^M - x_j^M \right\|)} \right)^{1/2} \tag{1}$$

Here, $M$ is the dimension of the reconstructed manifold, and $\Theta$ is the Heaviside function. If variable $u$ is independent of variable $x$, its conditional dispersion $\sigma_{xu}^{M}(\varepsilon)$ does not depend upon $\varepsilon$. If variable $x$ is the cause of $u$-variability, then conditional dispersion of the variable $u$ will decline for diminishing $\varepsilon$. As an illustration, we show in Fig.1 the conditional dispersion of two variables $x$ and $u$ of coupled Henon (1976) maps:

$$\begin{cases} x_{n+1} = 1 + y_n - 1.4 x_n x_n & \text{(2)} \\ y_{n+1} = 0.3 x_n + \alpha(v_n - y_n) & \text{(3)} \end{cases}$$

$$\begin{cases} u_{n+1} = 1 + v_n - 1.4 u_n u_n & \text{(4)} \\ v_{n+1} = 0.3 u_n + \beta(y_n - v_n) & \text{(5)} \end{cases}$$

Here variables $u$ and $x$ belong to two dynamical subsystems, (2) - (3) and (4) - (5). The interdependence of these subsystems is defined by coefficients $\alpha$ and $\beta$. When the connection is one-directional (for example, $\alpha=0$, $\beta=0.3$), i.e., $x$ is the cause of $u$ but is independent of $u$, the conditional dispersion of the $x$-variable does not depend on $\varepsilon$ (where $\varepsilon$ is the distance between synchronous points of $u$) but conditional dispersion of the $u$-variable declines for diminishing $\varepsilon$ (where $\varepsilon$ is the distance between synchronous points of $x$). When the connection is two-directional (for example, $\alpha=0.1$, $\beta=0.3$), the conditional dispersion of both variables declines for smaller $\varepsilon$, but a variable which provides a stronger causal force (i.e., $x$) has a dispersion with a less articulated slope. When variables are equally interdependent (i.e., synchronized), the conditional dispersions of both variables may have the same slope.

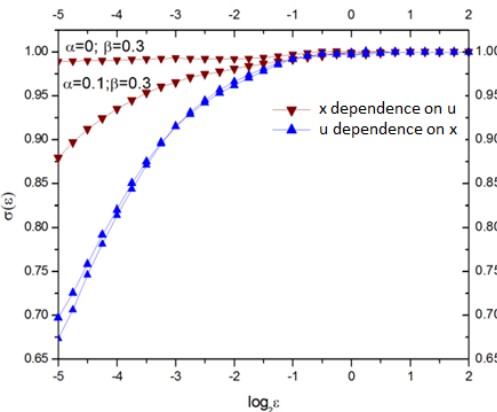

**Fig. 1.** The conditional dispersion of coupled dynamical variables $u$ and $x$ as described by equations (2) - (5). When connection is one-directional ($\alpha=0$, $\beta=0.3$), i.e., $x$ is the cause of $u$ but is independent of $u$, the conditional dispersion of the $x$-variable does not depend on $\varepsilon$, but conditional dispersion of the $u$-variable declines for diminishing $\varepsilon$. When the connection is two-directional ($\alpha=0.1$, $\beta=0.3$), the conditional dispersion of both variables declines for smaller $\varepsilon$, but the $x$-variable, which provides a stronger causal force, has a dispersion with weaker slope.

The results presented in Fig. 1 are based on a 4,000 data point calculations. If we reduce number of data points to ~ 150, the results qualitatively remain the same.

### 3. The case study of global warming causality

We will now employ the MCD approach in three numerical experiments for time series of atmospheric carbon dioxide concentration and surface temperature obtained from both direct instrumental measurements and model simulations. Since we now advance from a discrete attractor to measured and simulated time series, some assumptions need to be articulated. Indeed, despite the fact that numerous methods have been developed to better determine an embedding dimension (e.g., Abarbanel et al., 1993), it is still a challenge to determine embedding from a measured variable (such as temperature) because time series always have limited length and are corrupted with noise that can be misinterpreted as a higher dimension. We will treat the climate variables the same way as Hénon attractor variables (with evaluation "à la" Takens embedding, dimension 7). Fortunately, as it has been shown by Čenys et al. (1991), the MCD method is not very sensitive to the embedding dimension, and the slope of $\sigma(\varepsilon)$ curves increases only slightly with the increase of the dimension. We will use a hypothesis that Northern Hemisphere temperature is an observable of the global climate system, and the $CO_2$ concentration is an observable of the system of external forcing. An observable may not necessary have a straightforward connection to ("hidden") physical variables of the underlying system. The embedding theorem (e.g. Sauer et al., 1991) states that reconstructed space is topologically equivalent to the underlying system in the sense that there exists a continuous differentiable transform from reconstructed to hidden space.

### 3.1 Detecting causality in instrumental measurements.

First, we investigate the causal relationship between GISTEMP (Hansen *et al.*, 2006) surface temperature assessments for the Northern Hemisphere and atmospheric carbon dioxide concentration measurements (CO2 NASA GISS Data, 2016) spanning 1880 through 2016. For this purpose, we normalize the $CO_2$ and temperature time series by subtracting their mean values and by dividing over the standard deviation; we then calculate conditional dispersion of Northern Hemisphere temperature variability (as a function of distance $\varepsilon$ between synchronous points of the carbon dioxide time series) and conditional dispersion of carbon dioxide (as a function of distance $\varepsilon$ between synchronous points of the temperature time series). Any trends present in the data are preserved so as to avoid needless additional assumptions regarding the nature and origin of these trends.

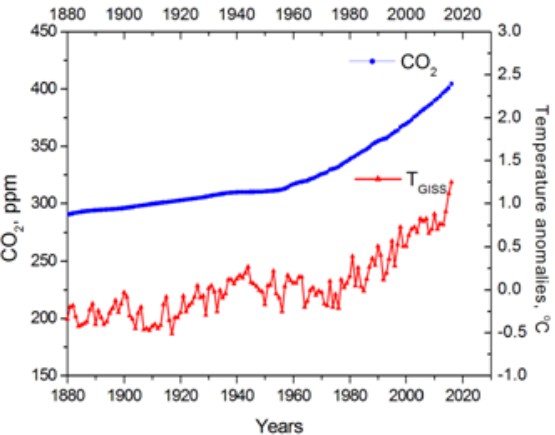 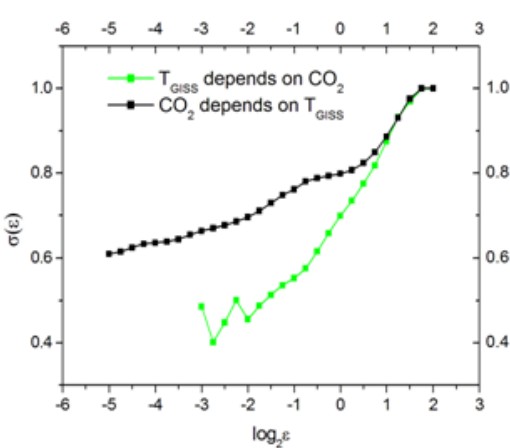

**Fig. 2. Left:** GISTEMP (Hansen *et al.*, 2006) surface temperature anomalies and atmospheric carbon dioxide concentration measurements (CO2 NASA GISS Data, 2016); **Right:** Conditional dispersion of instrumental measurements. Black curve is conditional dispersion of the carbon dioxide concentration. Green curve represents conditional dispersion of Northern Hemisphere temperature anomalies; its dependence on $\varepsilon$ is much stronger than that of the black curve, indicating that carbon dioxide is the cause of temperature changes.

It can be seen in Figure 2 that surface temperature and carbon dioxide are interdependent systems (the conditional dispersions of both variables depend upon $\varepsilon$). Nevertheless, *carbon dioxide is the causal force of global warming* because the dependence of the temperature conditional dispersion upon $\varepsilon$ is much stronger than the same

dependence of carbon dioxide conditional dispersion. When interpreting MCD results it is important to remember that we are dealing with relatively short time series that contain strong linear trends. We will show in Section 4 that the $\sigma(\varepsilon)$ slope can deviate significantly from the horizontal line because of linear correlation introduced by a trend, even in the case of completely independent time series. Therefore, it is not the absolute value of a slope but, instead, the difference (the "distance") between two slopes that speaks about the direction of causality.

### 3.2 Detecting causality in model simulations. Anthropogenic and natural (volcanic and solar) forcing.

We now apply MCD to the model simulations adopted from the Coupled Model Intercomparison Project Phase 5 (CMIP5) historical simulation experiments (Stocker, 2014). Estimates of the total forced component of Northern Hemisphere mean temperature have been derived by averaging over the full ensemble of CMIP5 multimodel all-forcing historical experiments (Mann *et al.,* 2014, Steinman *et al.*, 2015, Mann *et al.*, 2017). We generated 50 temperature series surrogates using a Monte Carlo resampling approach of Mann *et al.* (2017) and calculated conditional dispersion of Northern Hemisphere temperature variability for each of the 50 surrogates (as a function of distance $\varepsilon$ between synchronous points of carbon dioxide time series) and conditional dispersion of carbon dioxide (as a function of distance $\varepsilon$ between synchronous points of every surrogate time series). In all experiments we used the same atmospheric carbon dioxide concentration measurements (CO2 NASA GISS Data, 2016). We assume therefore that the effect of the surface temperature on $CO_2$ concentration has been naturally included in the $CO_2$ time series.

In Fig.3 it can be seen that the behavior of dispersions derived from multiple simulations' surrogates is quantitatively close to the dispersions obtained from the direct measurements, and therefore that carbon dioxide is the driver of temperature changes in the model simulations. Though in this experiment we applied MCD-testing network to the full ensemble of CMIP5 models, the same procedure can be applied to any sub-ensemble or to individual models if the task is to identify the models that are more consistent with the instrumental data in terms of causality.

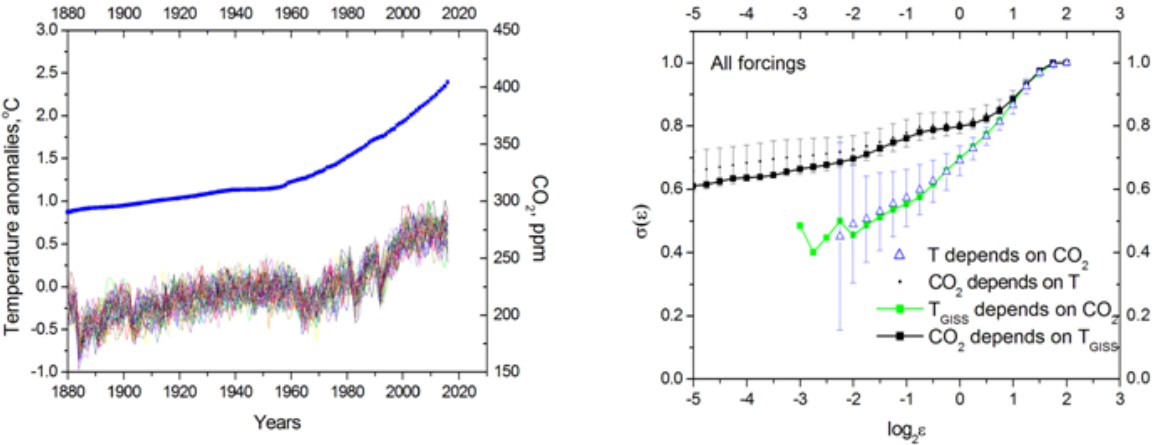

**Fig. 3.** Detecting causality in model simulations. Anthropogenic and natural (volcanic and solar) forcing**.** **Left**: Surrogates of model temperature deviations induced by both natural and anthropogenic forcing together with carbon dioxide concentration measurements (CO2 NASA GISS Data, 2016). **Right**: Conditional dispersion of the Northern Hemisphere temperature and carbon dioxide concentration. Black curve is conditional dispersion of the carbon dioxide concentration instrumental measurements; green curve represents conditional dispersion of the Northern Hemisphere temperature measurements (same as in the right panel of Fig.2). Blue triangles are mean of 50 multimodel surrogates' conditional dispersions of the Northern Hemisphere temperature; small black dots are mean of 50 multimodel surrogates conditional dispersions of carbon dioxide. Bars represent doubled standard deviation.

### 3.3 Detecting causality in model simulations. Anthropogenic forcing only.

We repeat the analysis described in paragraph 3.2 but for a separate ensemble of anthropogenic-only forcing experiments (Stocker, 2014, Mann *et al*., 2016 a, 2016 b, 2017).

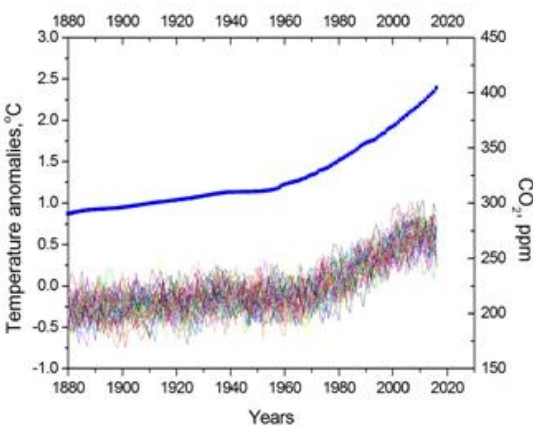 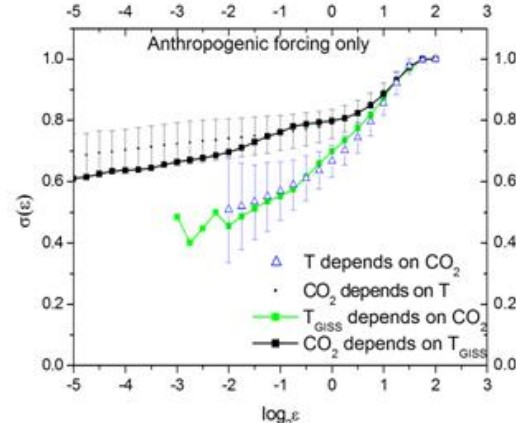

**Fig. 4.** Same as in Fig.3 but for anthropogenic CO$_2$ forcing only.

Interestingly, the results of the analysis change minimally when natural forcing (volcanic and solar) are excluded
(Fig.4), which implies the dominant causality role of carbon dioxide.

### 4.  Testing boundaries of MCD applicability.
Initially, the MCD approach was applied to causality measurements between deterministic chaotic time series
(Čenys *et al*., 1991). In this study, we expand its applicability to a situation where one of the time series (CO$_2$) is
essentially a regular trend and the history of observations for both CO$_2$ and temperature is relatively short. In the
next experiment we will test boundaries of MCD applicability and investigate if the MCD approach can distinguish
between *interdependent* processes, like CO$_2$ and temperature, and *independent* but highly autocorrelated processes.
For this purpose, we calculate conditional dispersion for two independent but highly autocorrelated time series
resembling properties of carbon dioxide series and temperature surrogates. For the carbon dioxide "role" we selected
a linear trend. GISS temperature surrogates were replaced by 50 red noise surrogates. Results of the conditional
dispersion calculations are shown in Fig.5.

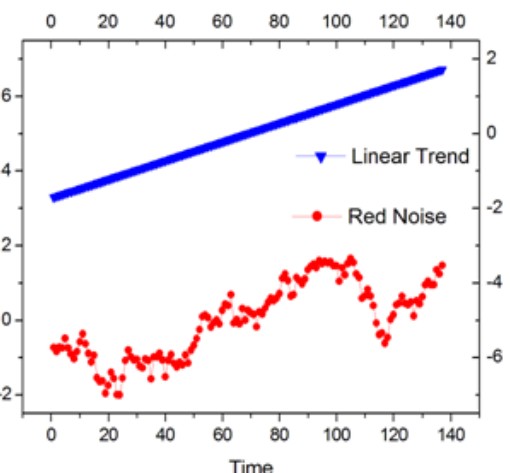 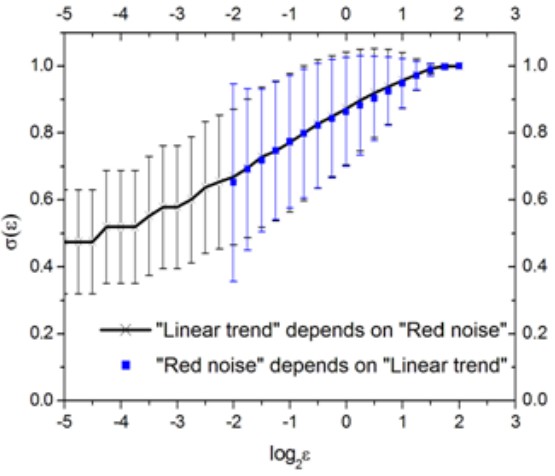

**Fig. 5.** Conditional dispersion of two independent processes having high autocorrelations. **Left:** Example of one-lag
autocorrelated (0.92) red noise series simulating statistical properties of GISS temperature record and a linear trend;
**Right:** Average conditional dispersion of 50 red noise surrogates. Error bars mark doubled standard deviation.

This example shows that, for relatively short time series, MCD is unable to discriminate cases of independence and
very strong interdependence (i.e., synchronization) because spontaneous local correlations may be induced with
autocorrelated red noise, leading to the same slope of conditional dispersions for both time series. Nevertheless,
unlike natural (CO$_2$ and temperature) time series, these correlations are not able to induce any directional causality.
In other words, were temperature and CO$_2$ equally interdependent, MCD would not be able to distinguish this

situation from independent trends and red noise. In reality though, both on the Quaternary and historical (i.e. current climate change) timescale, $CO_2$ and temperature display directional (albeit, different) causality. Specifically, temperature leads carbon dioxide on the orbital time scales (e.g., Van Nes *et al*., 2015), but, as we have demonstrated above, carbon dioxide is causally implicated for contemporary warming. This directional causality cannot be replicated with a trend plus red noise, confirming that the results presented in Figures 2-4 are not artifacts of noise.

**5. Conclusions.**

In this study we propose an additional climate model validation procedure that assesses whether causality signals between model drivers and responses are consistent with those observed in nature. Specifically, we suggest the method of conditional dispersion (MCD) as the best approach to directly measure the causality between model forcing and response. As an illustration of MCD applicability, we detect the causality signal between atmospheric carbon dioxide concentration and variations of global temperature. Our results suggest that there is a strong causal signal from the carbon dioxide series to the global temperature series or, in other words, that carbon dioxide is the principal *cause* of global warming. This conclusion is applicable to both direct instrumental measurements and multimodel temperature series surrogates. The strength of the causality signal does not significantly change when the additional contribution from natural factors (such as solar and volcanic) are accounted for, implying that increases in carbon dioxide are the main driver of observed warming. It is noteworthy that the causality between carbon dioxide and temperature anomalies is a directional causality: carbon dioxide affects temperature stronger than temperature affects carbon dioxide. This directional connection cannot be replicated using simplistic statistical models for the observed temperature increase (an independent trend and red noise).

Indeed, only laws of physics may identify the mechanism of causality, and therefore the causality is encoded in the differential equations of the mathematical models. Unfortunately, high uncertainty in natural forcing (e.g., Egorova et al., 2018) may be amplified by model uncertainties (e.g., Meehl et al., 2009), and despite the fact that multiple methods exist to detect causality in the data, none is perfect for the analysis of complex systems such as Earth's climate (McCracken, 2016).Therefore, a properly calibrated causality detection method like MCD, despite its simplicity (i.e., its basis in dynamical-systems theory),  may help to reduce these uncertainties in quantifying the climate response to different forcings by providing new data driven constraints. With our calculations, we calibrate MCD against existing measurements and simulations. As long as MCD is trusted as an insightful approach, it can be used for express testing of new models and, perhaps more importantly, can serve as a first test for any new external forcing candidate that may be considered as an alternative or supplement to $CO_2$.

**Code and data availability.** The MatLab source code and data (Verbitsky et al., 2019) are available at https://zenodo.org/record/2605142#.XJirxyIzbcs  (http://doi.org/10.5281/zenodo.2605142). Scripts were tested under MatLab version R2015b (last access: 25 March 2019).

**Author contributions.** MV conceived the research. MV and DV wrote the manuscript. All authors contributed equally to the design of the research and to editing the manuscript.

**Competing interests.** The authors declare that they have no conflict of interest.

**Acknowledgements.** Dmitry M. Volobuev is funded in part by the Russian Foundation for Basic Research, grant 19-02-00088-a. We are grateful to our two anonymous reviewers for their helpful comments.

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
