# Peer review of "Detecting causality signal in instrumental measurements and climate model simulations: global warming case study"

_Geoscientific Model Development, 2019_

## Referee Comment (RC1) · Anonymous Referee #1 · 19 Apr 2019

This paper presents a method to detect causality in climate time series. This method is based on principles of recurrences in dynamical systems. While interesting a priori, many points need to be improved and/or discussed.

Major points The literature review does not seem complete and misses crucial contributions. The paper is about attribution (of climate change), through the identification of causal links. There is an ample published literature on the subject, including in climate sciences. For instance, google scholar shows:

Hannart, A., Pearl, J., Otto, F. E. L., Naveau, P., & Ghil, M. (2016). Causal counter-factual theory for the attribution of weather and climate-related events. Bulletin of the

American Meteorological Society, 97(1), 99-110. (J. Pearl, who made the theory of causality, is a co-author of that paper)

Runge, J., Petoukhov, V., Donges, J. F., Hlinka, J., Jajcay, N., Vejmelka, M., ... & Kurths, J. (2015). Identifying causal gateways and mediators in complex spatio-temporal systems. Nature communications, 6, 8502.

A report of the US Academies of Science (National Academies of Sciences Engineering and Medicine 2016) also mention causality.

Although I do like dynamical systems, the transition from a 2-D discrete Hénon attractor to a "real world" problem sounds like a leap of faith. There are many numerical problems with the application of embedding methods ("à la" Takens). The main one is that there is no bound to the necessary embedding dimension, so that the low dimensional example that is treated is not sufficient to be convincing. The authors never mention questions linked to the so called "curse of dimensionality" to treat causality. When they treat the Hénon attractor, they use variables of the dynamical system, and do not need to evaluate embedding to make reconstructions. The climate application uses observables of the climate system (northern hemisphere temperature and CO2), which might not give a straightforward connection to variables of the underlying system. Therefore all interpretations might be misleading.

In Eqs (1-5), the systems have dimensionless variables, so that the choice of the range for epsilon is easy. The normalization of CO2 and temperature for figure 2, to compute Eq. (1) is not explained. The authors do not explain how they embed the climate time series. Their results are not reproducible from the text and figures.

I do not quite agree with the interpretation of Fig. 2b. The slopes of sigma(epsilon) are significantly positive for both ways (T and CO2). Therefore both observable interact with each other, in rather well physically understood way. The discussion on the slope to define the strength of the unidirectional interaction is irrelevant because it depends on the units of the variables and the shape of their probability distribution (from eye-

GMDD
balling Eq. (1)). The authors should compare how curves depart from a horizontal line when dealing with heterogeneous variables. If this type of analysis was done between a proxy for solar activity and temperature, I would expect a horizontal line. Is it the case?

What is the added value of the MCD analysis over the CMIP5 simulations and all the literature on attribution? The trend in observations cannot be obtained with control simulations and simulations with natural forcings. Only simulations with increasing CO2 can reproduce the recent trend. The analysis of this manuscript "just" reflects this known result. Such diagnostics (either visual, as reported by the IPCC or statistical in this manuscript) are relevant to measure causality. They do not state by which mechanism this causality operates: only first principles of physics can do that!

Specific points

Fig. 1 caption: alpha=0 and beta=0.3 is when x is the cause of u. The legend says the opposite (x depends on u). Please clarify or correct.

References National Academies of Sciences Engineering and Medicine (ed) (2016) Attribution of Extreme Weather Events in the Context of Climate Change. The National Academies Press, Washington, DC GMDD

---

## Referee Comment (RC2) · Anonymous Referee #2 · 10 May 2019

This paper argues that by using method of conditional dispersion (MCD) it is possible to detect causal relationship between surface temperature anomalies and $CO_2$ concentrations in climatological time series (observed and simulated). The method was initially developed to estimate interrelation in low-dimensional dynamical systems and applying it to climatological time series in indeed a novel approach withing the scope of this journal.

Two points need to be clarified to better understand results presented in the paper.

1. The transition from analyzing chaotic time series produced by two coupled Henon maps (briefly discussed in paragraph 2) to studying climatological time series is not as

straightforward as it is made in the paper. In particular, it requires some underlying assumptions about the nature of climatological system and interdependences between its variables which are not clearly mentioned in the paper.

2. In paragraphs 3.2 and 3.3, MCD is applied to a set of model simulations. It is unclear from the paper, if these models include parametrization of the effect of surface temperature on $CO_2$ emissions. If not, it would make the connection one-directional (see discussion in paragraph 2) which by itself explains the results presented in Figures 3 and 4.
* * *

---

## Author Comment (AC1) · 21 May 2019

Comment: This paper presents a method to detect causality in climate time series. This method is based on principles of recurrences in dynamical systems. While interesting a priori, many points need to be improved and/or discussed.

Answer: Dear Anonymous Referee #1, Thank you for your insightful review. We appreciate that you consider our approach to be interesting. The following is our response to your comments:

Major points

Comment: The literature review does not seem complete and misses crucial contributions. The paper is about attribution (of climate change), through the identification of causal links. There is an ample published literature on the subject, including in climate sciences. For instance, google scholar shows: Hannart, A., Pearl, J., Otto, F. E. L., Naveau, P., & Ghil, M. (2016). Causal counterfactual theory for the attribution of weather and climate-related events. Bulletin of the American Meteorological Society, 97(1), 99-110. (J. Pearl, who made the theory of causality, is a co-author of that paper); Runge, J., Petoukhov, V., Donges, J. F., Hlinka, J., Jajcay, N., Vejmelka, M., ... & Kurths, J. (2015). Identifying causal gateways and mediators in complex spatiotemporal systems. Nature communications, 6, 8502; A report of the US Academies of Science (National Academies of Sciences Engineering and Medicine 2016) also mention causality.

Answer: Though we have never aspired that our contribution may serve our readers as a review paper, we concede that you are correct and the reference list must be extended.

Action: In addition to a few items you advised us about, we will discuss and reference additional sources such as,

Abarbanel, H. D., Brown, R., Sidorowich, J. J., and Tsimring, L. S.: The analysis of observed chaotic data in physical systems. Reviews of modern physics, 65(4), 1331, 1993;

Attanasio, A., Pasini, A., and Triacca, U.: A contribution to attribution of recent global warming by out of sample Granger causality analysis. Atmospheric Science Letters, 13(1), 67-72, 2012;

Egorova, T., Schmutz, W., Rozanov, E., Shapiro, A.I., Usoskin, I., Beer, J., Tagirov, R., and Peter, T.: Revised historical solar irradiance forcing, Astron. Astrophys.,615, id A85. doi 10.1051/0004-6361/201731199, 2018;

McCracken, J. M.: Exploratory Causal Analysis with Time Series Data. Synthesis Lectures on Data Mining and Knowledge Discovery, 8(1), 1-147, 2016;

Meehl, G. A., Arblaster, J. M., Matthes, K., Sassi, F., and van Loon, H.: Amplifying the Pacific climate system response to a small 11-year solar cycle forcing. Science, 325(5944), 1114-1118, 2009;

O'Brien, J. P., O'Brien, T. A., Patricola, C. M., and Wang, S. Y. S.: Metrics for understanding large-scale controls of multivariate temperature and precipitation variability. Climate Dynamics, 1-19, 2019;

Sauer, T., Yorke, J. A., and Casdagli, M.: Embedology. Journal of statistical Physics, 65(3-4), 579-616, 1991;

Vejmelka, M. et al.: Non-random correlation structures and dimensionality reduction in multivariate climate data. Climate Dyn. 44, 2663–2682, 2015;

Comment: Although I do like dynamical systems, the transition from a 2-D discrete Hénon attractor to a "real world" problem sounds like a leap of faith. There are many numerical problems with the application of embedding methods ("à la" Takens). The main one is that there is no bound to the necessary embedding dimension, so that the low dimensional example that is treated is not sufficient to be convincing. The authors never mention questions linked to the so called "curse of dimensionality" to treat causality. When they treat the Hénon attractor, they use variables of the dynamical system, and do not need to evaluate embedding to make reconstructions. The climate application uses observables of the climate system (northern hemisphere temperature and CO2), which might not give a straightforward connection to variables of the underlying system. Therefore all interpretations might be misleading.

Answer: We use Hénon attractor just as an illustration of the MCD concept because we want our readers to be well equipped before they review and interpret the sigma(epsilon) curves that are presented in the following chapters. Nevertheless, your

СЗ

concern regarding the "curse of dimensionality" is legitimate. Yes, we treat the climate variables the same way as Hénon attractor variables (with evaluation "à la" Takens embedding, dimension 7). Fortunately, though, as it has been shown by Čenys et al. (1991), the MCD method is not very sensitive to the embedding dimension and the slope of sigma(epsilon) curves increases only slightly with the increase of the dimension. Indeed, despite the fact that numerous methods have been developed to better determine an embedding dimension (e.g., Abarbanel et al. 1993), it is still a challenge to determine embedding from a measured variable (such as temperature) because time series always have limited length and are corrupted with the noise which can be misinterpreted as a higher dimension. We use a hypothesis that NH temperature is an observable of the global climate system and CO2 concentration is an observable of the system of external forcing. An observable may not necessary have straightforward connection to ("hidden") physical variables of the underlying system. The embedding theorem (e.g. Sauer et al., 1991) states that reconstructed space is topologically equivalent to the underlying system in a sense that there exists a continuous differentiable transform from a reconstructed to the hidden space.

Action: We will add this discussion to the appropriate parts of the paper.

Comment: In Eqs (1-5), the systems have dimensionless variables, so that the choice of the range for epsilon is easy. The normalization of CO2 and temperature for figure 2, to compute Eq. (1) is not explained. The authors do not explain how they embed the climate time series. Their results are not reproducible from the text and figures.

Answer: CO2 and temperature time series have been normalized by subtracting their mean values and by dividing over standard deviation.

Action: This will be added to the text

Comment: I do not quite agree with the interpretation of Fig. 2b. The slopes of sigma(epsilon) are significantly positive for both ways (T and CO2). Therefore both observable interact with each other, in rather well physically understood way. The discussion on the slope to define the strength of the unidirectional interaction is irrelevant because it depends on the units of the variables and the shape of their probability distribution (from eye-balling Eq. (1)). The authors should compare how curves depart from a horizontal line when dealing with heterogeneous variables. If this type of analysis was done between a proxy for solar activity and temperature, I would expect a horizontal line. Is it the case?

Answer: It is not the case. We have relatively short time series with a strong linear trend. We show in Fig. 5 that the sigma(epsilon) slope can deviate significantly from the horizontal line because of linear correlation introduced by a trend, even in the case of completely independent time series. Solar activity does have a trend and the corresponding sigma(epsilon) curve will deviate from the horizontal line. Therefore, in our paper, we are focusing not on the slope by itself but on the difference between two slopes, both in instrumental measurements and in model simulations.

Action: We will articulate this notion more clearly in the text.

Comment: What is the added value of the MCD analysis over the CMIP5 simulations and all the literature on attribution? The trend in observations cannot be obtained with control simulations and simulations with natural forcings. Only simulations with increasing CO2 can reproduce the recent trend. The analysis of this manuscript "just" reflects this known result. Such diagnostics (either visual, as reported by the IPCC or statistical in this manuscript) are relevant to measure causality. They do not state by which mechanism this causality operates: only first principles of physics can do that!

Answer: We can't agree more that only laws of physics may identify the mechanism of causality. In this sense the causality is encoded in the differential equations of the mathematical models. With our calculations, we are not challenging the consensus whether CO2 is the cause of the temperature increase, but rather calibrate MCD against existing measurements and simulations. Unfortunately, despite the fact that multiple methods exist to detect causality in the data, none is perfect for analysis of complex systems

such as the Earth climate (McCracken, 2016). High uncertainty in natural forcing (e.g., Egorova et al., 2018) may be amplified by model uncertainties (e.g. Meehl et al., 2009). Therefore, a properly calibrated causality detection method like MCD may help to reduce these uncertainties in quantifying the climate response to different forcings by providing new data driven constraints. As long as MCD is recognized as a trusted approach, it can be used for express testing of new models and, may be more importantly, can serve as a first test for any new candidate external forcing that may be considered as an alternative or supplement to CO2.

Action: We will add this discussion to the test.

Comment: Specific points Fig. 1 caption: alpha=0 and beta=0.3 is when x is the cause of u. The legend says the opposite (x depends on u). Please clarify or correct.

Answer: We agree that it may be confusing

Action: We will make a more clear legend.

---

## Author Comment (AC2) · 21 May 2019

Comment: This paper argues that by using method of conditional dispersion (MCD) it is possible to detect causal relationship between surface temperature anomalies and CO2 concentrations in climatological time series (observed and simulated). The method was initially developed to estimate interrelation in low-dimensional dynamical systems and applying it to climatological time series is indeed a novel approach within the scope of this journal.

Answer: Dear Anonymous Referee #2, Thank you for your review and suggestions. We appreciate that you consider our approach to be novel and appropriate for Geoscientific
Model Development. The following is our response to your comments:

Comment: Two points need to be clarified to better understand results presented in the paper

The transition from analyzing chaotic time series produced by two coupled Henon maps (briefly discussed in paragraph 2) to studying climatological time series is not as straightforward as it is made in the paper. In particular, it requires some underlying assumptions about the nature of climatological system and interdependences between its variables which are not clearly mentioned in the paper.

Answer: We use a hypothesis that NH temperature is an observable of the global climate system and CO2 concentration is an observable of the system of external forcing. An observable may not necessary have straightforward connection to ("hidden") physical variables of the underlying system. The embedding theorem (e.g. Sauer et al., 1991) states that reconstructed space is topologically equivalent to the underlying system in a sense that there exists a continuous differentiable transform from a reconstructed to the hidden space.

Reference: Sauer, T., Yorke, J. A., and Casdagli, M.: Embedology. Journal of statistical Physics, 65(3-4), 579-616, 1991

Action: We plan to add this discussion to the paper.

Comment: In paragraphs 3.2 and 3.3, MCD is applied to a set of model simulations. It is unclear from the paper, if these models include parametrization of the effect of surface temperature on CO2 emissions. If not, it would make the connection one-directional (see discussion in paragraph 2) which by itself explains the results presented in Figures 3 and 4.

Answer: We agree with your comment that clarification about the origin of CO2 time series used to produce Figures 3 and 4 is needed. In fact, in all experiments we used the same atmospheric carbon dioxide concentration measurements (CO2 NASA GISS

GMDD
Data, 2016). We assume therefore that the effect of the surface temperature on CO2 concentration has been naturally included in the CO2 time series.

Reference: CO2 NASA GISS Data: https://data.giss.nasa.gov/modelforce/ghgases/Fig1A.ext.txt, 2012

Action: We will clarify this in the text.

GMDD

---

## Author Response (AR2)

**Response to Anonymous Referee #1**

Dear Anonymous Referee #1,

We appreciate very much your additional questions. They force us to be more articulate in our interpretation of the results and therefore make our paper more useful for potential readers.

**Comment:** I am still dubious on the interpretation of figures 3-5. The comparison with a Hénon map makes the hypothesis that the two analyzed variables have the same scale (the authors normalize them) AND the same equations or dynamics. The other example they take uses a linear trend and red noise (therefore different dynamics) for which they cannot conclude anything. They state that "In reality though, carbon dioxide affects temperature stronger than temperature affects $CO_2$, and this situation cannot be replicated by a trend and red noise". This is rather obscure. What is the rationale for this comparison, which is certainly wrong on quaternary time scales?
**Answer:** We agree that the sentence you quote is not clear. Instead, we have to say that both on the Quaternary and historical (i.e. current climate change) timescale, $CO_2$ and temperature display directional (albeit, different) causality. Specifically, temperature leads carbon dioxide on the orbital time scales (e.g. Van Nes et al., 2015), but, as we have demonstrated, carbon dioxide is causally implicated for contemporary warming. This directional causality cannot be replicated with a trend plus red noise, confirming that the results presented on Figures 2-4 are not artifacts of noise.
**Action:** We will clarify this in the text. **DONE: p.8 lines 1-6**

**Comment:** I grant that those are points of discussion, but I feel that the dynamical system illustration (or any other form of idealized system) lacks complexity, like coupling two *different* systems, for a more useful interpretation of the figures.
**Answer:** This is very interesting and deeply philosophical question. We will, probably, all agree that any mathematical model is a simplification of reality. Otherwise, the interpretation of model's results would be as difficult as interpretation of the nature. Then, how simple should a model be? At this point, the science of mathematical modeling becomes an "art" of mathematical modeling, and a lot of arguments can be made in favor or against complexity. For example, some valid arguments have been made that in many situations the dynamical models can provide just as much insight as more complicated models (Saltzman, 1990). We do not aspire, at this point, to provide a definitive answer to this discussion, but we see a value in a complementing three-dimensional modeling approach with a relatively simple test based on dynamical system vision.

Saltzman, B.: Three basic problems of paleoclimatic modeling: A personal perspective and review. Climate Dynamics, 5(2), 67-78, 1990
**Action:** We will add some discussion to this point. **DONE: p.8 lines 26- 27**

**Comment:** I still do not understand how (non) stationarity is treated in the climate data.
**Answer:** Only minimal pre-processing of the data (normalizing) has been made. Any trends present in the data are preserved so as to avoid needless additional assumptions regarding the nature and origin of these trends.
**Action:** We will mention it in the text. **DONE: p.5 lines 27-28**

**Comment:** The authors do not state how many points they simulate on the Hénon attractor. If they only take ~140 points, would they get similar results?
**Answer:** The results presented on Fig. 1 are based on 4,000-points calculations. If we use instead only 140 points, qualitatively results remain the same. If we use an ensemble of 140-points data, the average results become identical to those presented on Fig. 1. Accordingly, when we deal with the model data (Figs. 3 and 4), we employ a 50-surrogates ensemble of the data to estimate the statistical deviation of the results.
**Action:** We will make this clarification in the text. **DONE: p.5 lines 1-2**

[revised manuscript text omitted]